# The Association of R-Loop Binding Proteins Subtypes with CIN Implicates Therapeutic Strategies in Colorectal Cancer

**DOI:** 10.3390/cancers14225607

**Published:** 2022-11-15

**Authors:** Wenchao Zhao, Qian Pei, Yongwei Zhu, Dongdong Zhan, Guo Mao, Meng Wang, Yanfang Qiu, Ke Zuo, Haiping Pei, Lun-Quan Sun, Ming Wen, Rong Tan

**Affiliations:** 1General Surgery Department, Xiangya Hospital, Central South University, Changsha 410008, China; 2Xiangya Cancer Center, Xiangya Hospital, Central South University, Changsha 410008, China; 3Key Laboratory of Molecular Radiation Oncology Hunan Province, Changsha 410008, China; 4National Clinical Research Center for Geriatric Disorders, Xiangya Hospital, Central South University, Changsha 410008, China; 5Department of Neurosurgery, Xiangya Hospital, Central South University, Changsha 410008, China; 6Hunan International Scientific and Technological Cooperation Base of Brain Tumor Research, Xiangya Hospital, Central South University, Changsha 410008, China; 7State Key Laboratory of Proteomics, Beijing Proteome Research Center, National Center for Protein Sciences (Beijing), Beijing Institute of Lifeomics, Beijing 102206, China; 8Science and Technology on Parallel and Distributed Processing Laboratory, National University of Defense Technology, Changsha 410073, China; 9Hunan International Science and Technology Collaboration Base of Precision Medicine for Cancer, Changsha 410008, China; 10Center for Molecular Imaging of Central South University, Xiangya Hospital, Changsha 410008, China; 11Hunan Key Laboratory of Aging Biology, Xiangya Hospital, Central South University, Changsha 410008, China

**Keywords:** chromosome instability, R-loop binding proteins, molecular subtype, drug sensitivity

## Abstract

**Simple Summary:**

R-Loops, finely regulated by R-Loop binding proteins (RLBPs), play pivotal roles in maintaining genomic stability. By integrated proteogenomic analysis, we identified two RLBPs subtypes with distinct prognostic and therapeutic differences in colorectal cancer (CRC). Cluster-I (CI), characterized by high expression of RLBPs, was associated with chromosomal instability (CIN), better prognosis, and sensitivity to drugs targeting genome integrity and EGFR. Cluster-II (CII), characterized by low expression of RLBPs, was associated with mucinous adenocarcinoma, right-sided colon cancer, and poor prognosis. High inflammatory signaling pathway and lymphocyte infiltration enriched in CII, indicating potential application of drugs targeting inflammatory and immune response. Our research might be helpful for the precision treatment of CRC.

**Abstract:**

Chromosomal instability (CIN) covers approximately 65 to 70% of colorectal cancer patients and plays an essential role in cancer progression. However, the molecular features and therapeutic strategies related to those patients are still controversial. R-loop binding proteins (RLBPs) exert significant roles in transcription and replication. Here, integrative colorectal cancer proteogenomic analysis identified two RLBPs subtypes correlated with distinct prognoses. Cluster I (CI), represented by high expression of RLBPs, was associated with the CIN phenotype. While Cluster II (CII) with the worst prognosis and low expression of RLBPs was composed of a high percentage of patients with mucinous adenocarcinoma or right-sided colon cancer. The molecular feature analysis revealed that the active RNA processing, ribosome synthesis, and aberrant DNA damage repair were shown in CI, a high inflammatory signaling pathway, and lymphocyte infiltration was enriched in CII. In addition, we revealed 42 tumor-associated RLBPs proteins. The CI with high expression of tumor-associated proteins was sensitive to drugs targeting genome integrity and EGFR in both cell and organoid models. Thus, our study unveils a significant molecular association of the CIN phenotype with RLBPs, and also provides a powerful resource for further functional exploration of RLBPs in cancer progression and therapeutic application.

## 1. Introduction

Chromosomal instability (CIN) and microsatellites instability (MSI) are two major characteristics of colorectal cancer (CRC) [1,2]. Despite substantial advanced in genomic features for cancer, the clinical significance of those features and potential mechanisms to guide therapeutic strategies have not yet been defined. The MSI in colon cancer is caused by deficiency or dysfunction of the mis-match repair pathway. The early diagnosis of MSI in CRC becomes the therapeutic standard for applications of fluorouracil and immune checkpoint inhibitors, while the molecular mechanisms underlying CIN and related therapeutic sensitivity are still controversial.

R-loop is processed by the transient cross-reaction of RNA with template DNA to form a DNA-RNA hybrid. The structures are prevalent in rDNA loci, telomeric regions, actively transcribed regions, and mitochondrial DNA [3,4,5,6]. Extensive physiological and pathological processes, such as immunoglobulin recombination, transcriptional regulation, DNA replication, telomere homeostasis, CpG island promoter methylation, and DNA lagging strand synthesis in activated B cells, have been associated with the formation and mediation of R-loop [5,7]. The R-loop structures cause DNA damage in certain contexts and are regarded as a major source of genomic instability [8]. Thus, homeostatic maintenance of R-loop plays an important role in genome stability [9].

R-loop homeostasis is regulated by a series of R-loop binding proteins (RLBPs). Thus, aberrant expression or dysfunction of R-loop binding proteins impairs genomic integrity and is associated with diverse diseases. Mutations in the DNA/RNA helicase senataxin (SETX) result in neurodegenerative disorders, ataxia with oculomotor apraxia type 2 and amyotrophic lateral sclerosis type 4 (AOA2/ALS4) [10]. The EWS-FIL1 fusion induces R-loop accumulation and perturbs homologous recombination repair, a process that results in Ewing sarcoma and underlies the mechanism of chemosensitivity [11]. Moreover, several factors that are involved in DNA damage repair have been demonstrated to be RLBPs. BRCA1 and the Fanconi anemia (FA) pathway factors FANCA, FANCD2, and TOP1 have been reported to bind R-loop during DNA damage repair [12,13,14,15]. Dysfunction of BRCA1 and FA factors is associated with breast cancer and Fanconi anemia, respectively. Due to the significant function of TOP1 in genome maintenance, it is also a desirable chemotherapeutic target. In this regard, advances in elucidating the expression and mechanistic signatures of RLBPs are essential for understanding the regulation of genome stability and cancer, further suggesting desirable therapeutic strategies.

Our study initiated a comprehensive proteomics analysis of 204 RLBPs in three colorectal (CR) proteogenomic databases. We revealed two RLBP clusters with distinct prognoses. The two clusters correlated with CIN phenotypes, anatomical location, and pathological status. Furthermore, the two clusters were distinguished by physiological features and tumor microenvironment (TME). In addition, we also identified 42 tumor-related RLBPs. We revealed that the clusters with high expression of tumor-related RLBPs displayed different drug sensitivity involved in EGFR and genome stability pathways. Thus, the associations of RLBPs with cancer provide a theoretical basis for further functional investigation of R-loop binding proteins in cancer. The RLBP clustering also bridges a gap between CIN and RLBPs, facilitating personalized diagnosis and appropriate selection for therapy in clinical application.

## 2. Results

### 2.1. Proteogenomic Databases for RLBP Analysis

A continuously increasing number of methods have been developed to identify RLBPs and establish R-loop-interacting protein databases. The Natalia Gromak group performed affinity purification using the S9.6 antibody to obtain 463 predicted R-loop binding proteins, and the Vivian G. Cheung group employed synthesized hybrids to carry out precipitation to obtain 803 R-loop binding proteins [16,17]. We thus employed a shared 197 overlapping RLBPs from the two databases (Appendix A). In addition, some well-established hybrid-binding proteins, such as RNase H1/2, SETX, AQR, and XPG, were lost due to methodological bias [10,18,19,20,21]. To define the extensive landscape of RLBPs and establish their relevance to colon cancer, we carried out proteogenomic analysis using 204 RLBPs, including the 197 detected RLBPs and 7 acknowledged RLBPs (Appendix A), in three colorectal cancer databases and CCLE colorectal cancer cell line databases [22,23,24,25,26,27,28]. The 204 RLBPs were used for unsupervised clustering analysis and signature identification. In addition, survival differences, clinical relevance, the immune microenvironment, somatic mutation status, genome stability, and drug sensitivity were analyzed.

### 2.2. Clustering of Colorectal Cancers Based on Proteomic Data of RLBPs

To further investigate RLBP expression patterns in tumor tissues, we performed nonnegative matrix factorization (NMF)-based unsupervised clustering using 204 RLBPs in the 2014 CPTAC CRC database [22] (Figure 1a and Appendix A). The tumor samples can be classified into two subtypes with distinct clinical outcomes. Cluster I (CI) had a better prognosis and Cluster II (CII) had a worse prognosis (Figure 1b and Appendix A). To further facilitate stratification, we selected 14 variable important genes (DDX21, MYBBP1A, GTPBP4, UTP14A, NOL6, DDX5, HSP90AB1, PRPF3, NOP2, SMARCA5, WDR36, PDCD11, and EMG1, MKI67) and established a clustering model using the random forest algorithm (Figure 1c and Appendix A). We also found the 14 variable important genes were enriched in malignant cells in CRC scRNA analyses (Figure 1d). We then used the model to classify 2019 CPTAC or 2020 Cancer Cell CRC databases. We matched the subtypes assigned by the random forest model to NMF subtypes in those two CRC databases and revealed that the accuracy was higher than 77%, indicating the reliability of the clustering method [23,24] (Appendix A). Based on the subtypes classified by NMF in 2014 or by clustering model in 2019 and 2020 databases, we noticed that the expressions of the RLBPs were consistently high in CI and low in CII (Figure 1e, Appendix A). To further validate the RLBPs expression in two clusters, we performed immunohistochemistry (IHC) of MCM3 and MCM5 in colorectal cancer tissues (Figure 1f). The results showed that the two genes exhibited consistent expression patterns in patients from two clusters. Indeed, the IHC scores of MCM3 and MCM5 were highly correlated, further supporting that the RLBPs could be markers to distinguish CRC patients. Thus, the CI and CII were subsequently referred to as the RLBP-proficient (CI-pRLBP) cluster and deficient RLBP (CII-dRLBP) cluster, respectively. Among all 204 RLBPs, 16 encoding genes showed cluster-dependent correlation with prognosis, as manifested by their significant impact on survival in CII, of note, high expression of 15 genes, including DDX21 and DKC1, led to worse outcomes in CII (Figure 1g–i and Appendix A). Thus, RLBP-based clustering revealed the significance of genes associated with tumors, which is frequently underestimated in the observation of total tumor patient cohorts.

### 2.3. Prognostic Significance of the RLBPs-Based Cluster

To gain the clinical relevance for RLBPs clusters, we analyzed clinical features related to two clusters. The two clusters were highly associated with cancer pathological types, tumor site, histological type, and the transcriptomic subtype in 2014 CPTCA CRC databases, while, we did not observe significant relevance of clusters with MSI status, sex, or lymphatic metastasis [22,23,29] (Figure 2a,b and Appendix A). The higher percentage of patients with rectum tumors (43.6%) or with left-sided colon cancer (67.2%) were distributed in CI-pRLBP (Figure 2c,d). While the CII-dRLBP contained more patients with right-sided colon cancer (54.3%) or mucinous adenocarcinoma (20%) (Figure 2d,e). Interestingly, we also noticed that CI was highly associated with the CIN phenotype (36.6%) compared to CII (29.0%) in 2014 CPTAC CRC databases. The same result was obtained in the 2019 CPTAC CRC databases, CI (64.8%) was highly enriched in CIN phenotype compared to CII (25.0%) (Figure 2f).

To examine whether RLBP clusters could serve as an independent prognostic factor, we employed univariate and multivariate cox regression models using clinical information, including age, gender, stage, residual tumor, and pretreatment CEA level. The result showed that the cluster was a robust and independent prognostic biomarker for evaluating CRC patient OS (Figure 2g,h; HR = 6.0, 95% CI 1.23–28.1, *p* < 0.05). Moreover, Cox models integrated clusters with stage, residual tumor, distant metastasis, and pretreatment CEA level greatly enhanced prognostic accuracy, implying that RLBPs clusters further facilitate prognostic prediction of CRC patients (Figure 2i). The results suggested that RLBPs-based clusters reflect the difference clinical features and predict the prognosis of CRC patients.

### 2.4. Different Molecular and Immune Characteristics of the RLBPs-Based Cluster

RLBPs are essential for genome maintenance; thus, we further compared the mutations between the two subtypes. However, we observed no apparent variations in the indel or point mutation burden between CI and CII in either the TCGA or CPTAC databases (Appendix A, see Section 4). While, we found the CI cluster showed higher average copy number variation (CNV) scores than CII (Figure 3a). Aberrant expression or dysfunction of DNA damage repair proteins implies inappropriate DNA damage repair, which finally results in genome instability [30]. We analyzed the expression of 276 DNA damage repair proteins in two clusters of three CRC databases. Interestingly, proteins involved in DNA damage repair showed apparently high expression in CI (Figure 3b and Appendix A). Furthermore, 12 DNA damage repair proteins, such as RFC1, RFC4, PAPR1, and PCNA, which engage in one or more specific DNA repair pathways, like translesion DNA synthesis (TLS), nucleotide excision repair (NER), base excision repair (BER), mismatch repair (MMR) and double-strand breaks (DSBs) repair, showed consistently higher expressions in CI than CII throughout the four proteomic cohorts (*p* < 0.05, Wilcoxon signed-rank test). It is notable that the expression of MSH2 showed elevation in CI in both CPTAC databases, while exhibiting the opposite result in the 2020 Cancer cell CRC database (Figure 3c).

RLBPs play important roles in transcription, translation, and RNA metabolism. However, unexpectedly, the phosphorylations of POLR2A at S2 and S5, which indicate transcriptional activity, exhibited no differences between CI and CII (Figure 3d and Appendix A). The further Gene Set Enrichment Analysis (GSEA) revealed higher RNA processing activity in CI, while considerable enhancement of inflammatory response in CII in all the three CRC databases (Figure 3e,f and Appendix A). These results suggested that two clusters engaged in different molecular signaling pathways and hold distinct DNA repair capabilities.

### 2.5. The Related Immune Microenvironments in CI and CII

The composition, infiltration, and activity of immune cells constitute the tumor immune microenvironment and confer antitumor activity. To investigate differential immune microenvironments in two subtypes, we employed several methods to profile distinct immune cells signatures mapping to the proteomic clusters [31,32,33,34]. Similar as molecular features, the CII showed consistently higher lymphocyte fraction, as evidenced by the immune scores, in both the 2014 and 2019 CPTAC CRC databases (Figure 4a and Appendix A), indicating extensive immune cell infiltration in the CII subtype compared to the CI subtype of cancer. Furthermore, in spite of different analysis methods, we found that macrophages were highly accumulated in CII in the 2014 CPTAC CRC database (Figure 4b and Appendix A). The immunochemical staining of macrophages in colorectal cancer tissues further emphasized the increased tumor-associated infiltration of macrophages in CII patients (Figure 4b and Appendix A) [34]. In the 2019 CPTAC CRC database, we also noticed that the higher percentage of monocytes, which is the precursor of macrophages, is presented in CII (Figure 4c). Six immune subtypes have been previously characterized based on TCGA transcriptome databases [35]. We noticed that CI patients enriched in immune subtypes C1 and C2. While CII patients spanned more board range of immune subtypes, especially accumulating in immune subtypes C3 and C4, which displayed inflammatory and macrophages signatures respectively (Figure 4d).

By analyzing the correlation between the RLBPs and immune cell populations, we found that the expression of DKC1 showed a negative correlation with CD8+ Tcm cells, especially in CII (Figure 4e,f and Appendix A). To further validate the correlation of DKC1 with CD8+ T cells, we collected colorectal cancer tissues and immunohistochemically stained the expression of CD8 and DKC1. We measured the correlation of DKC1 expression with CD8+ T cells at tumor sites, stromal sites, and the invasive front (where the tumor invaded the normal lamina propria). The results showed that DKC1 expression was negatively correlated with CD8+ T cells populations at the invasive front (Figure 4g,h), but not at tumor sites and stromal sites (Appendix A). The same results were found in macrophages (Appendix A). DKC1 expression was positively correlated with the macrophages marker, CD68, at the invasive front of colon cancer tumors (Appendix A). Meanwhile, we grouped the patients according to DKC1 expression and observed higher expression of CD8 (Figure 4i) or CD68 (Appendix A) in the DKC1-low group, which further demonstrated that RLBPs performed potential regulatory functions in the immune microenvironment. The immune modulators that are secreted by or located in immune cells and nonimmune cells play critical roles in orchestrating the immune environment. We interestingly found a majority of immune modulators were preferentially expressed in CII, further supporting the highly modulatory environment of immune cells in CII in both mRNA and protein levels (Figure 4j and Appendix A). In addition, the MHC II component (HLA-DPB1), TGFβ1, TNFRSF4, and TNFSF4 showed consistently high expressions in CII in both the 2014 and 2019 CRC databases. However, MHC I components exhibited no consistent differential expression in the two clusters (Figure 4k and Appendix A). These data indicated that two clusters have different degrees of infiltration by immune cell types and RLBPs could affect immune cells in TME.

### 2.6. A Total of 42 RLBPs Show Consistently High Expression in Tumor Tissues

To obtain the tumor high expression RLBPs, we compared the expressions of 204 RLBPs between tumor tissues and normal adjacent tissues (NATs) in two colorectal databases [22,23]. We obtained 101, and 65 proteins were highly expressed in tumor tissues in the 2014 and 2019 CPTAC colorectal cancer databases, respectively (*p* < 0.05, log_2_FC > 0.5) (Figure 5a). In total, 42 hybrid-binding proteins with consistently high expression were detected in the two proteomic databases and identified as tumor-associated hybrid-binding proteins (Figure 5b–d and Appendix A). In addition, MCM3 and IGF2BP3 exhibited consistent upregulation in tumor tissues compared to NATs in six multiple cancer proteomic databases [23,36,37,38,39,40] (*p* < 0.05, log_2_FC > 0.5) (Appendix A). To validate cell types that hold the high expression of those genes, we obtained eight major cell populations referring to fibroblast, epithelial cells, malignant cells, Innate lymphoid cells (ILC), CD4+ T cells, CD8+ T cells B cells, myeloid cells from single-cell RNA-seq (scRNA) datasets [41] (Figure 5e). The 42 RLBPs were highly expressed in a group of malignant cells which was a subgroup from the epithelial cell population, further implying that those proteins may play tumor-promoting roles [41] (Figure 5f). To further verify protein expression, we collected paired colon tumors and NATs and obtained that two of 42-tumor-associated RLBPs, MCM3, and MCM5, exhibited very high expression in tumors compared to the NATs (Figure 5g,h).

High expression of proteins can be affected by copy number alterations (CNA) at the genomic level or by transcriptional/translational regulation. We found high frequency of positive correlations of protein expression with mRNA expression and CNA and a negative correlation with methylation [27,42] (Figure 5i and Appendix A). We further summarized six genes (IGF2BP3, LYAR, RALY, STAU1, SYNCRIP, and TOP1) that displayed high correlations between mRNA and protein expression in both cancer and cell line databases (Figure 5j and Appendix A). The expression of 5 genes (PRKDC, RALY, STAU1, SYNCRIP, and TOP1) displayed a strong positive correlation with CNA (r > 0.3, *p* < 0.05) (Figure 5k). Notably, seven or eleven genes showed both significantly high gene-level amplifications and mRNA in cancer and cell line databases respectively. Furthermore, the expression of one gene, RALY, showed an intimate correlation with gene amplification, promotor methylation, and mRNA level in the cancer database. Unfortunately, we did not observe a consistent correlation between CpG promoter methylation and protein expression in the cancer and cell line databases (Figure 5l). These findings suggested that the expressions of RLBPs were consistently high in tumor tissues and cells may possess motivation oncogenesis in CRC.

### 2.7. Alternative Therapeutic Drugs Are Selected and Used to Treat RLBP-Expressing Tumors

To further evaluate the therapeutic potential of the 42 tumor-associated genes, we analyzed the CRISPR screening project-CERES scores across the colorectal cancer cell lines [25]. More than 45% of the genes (19/42) had poor therapeutic index, with a score lower than −1 (Appendix A). Moreover, 9 of the 42 tumor-associated genes were identified as core fitness genes across three datasets (Appendix A), implying that those genes are essential for physiological processes and thus have a low therapeutic index [25,27,43,44,45,46,47]. While, the metastatic features showed no apparent correlation with the CERES scores of RLBPs (Appendix A).

Alternative therapeutic strategies should be considered for tumors with high RLBP expression. Since we observed that 14 genes involved in subtypes classification were enriched in malignant tumor cells (Figure 1d), we divided CRC cancer cells into two clusters and observed the 42 gene expressions were accumulated in CI in cells, which was reminiscent of subtypes in cancer databases (Figure 1a and Figure 6a). We turned to a large-scale drug screening project, the GDSC1 dataset [48]. We interestingly found drugs involved in EGFR and genome stability were highly sensitive in CI compared to CII (Figure 6b and Appendix A). Pathway enrichment analyses showed CI cell cluster was enriched in oncogenic MAPK signaling and PI3K/AKT pathways, which were two critical downstream pathways for EGFR (Figure 6c). Moreover, DNA replication and DNA damage repair pathways were enriched in the CI cluster (Appendix A).

Furthermore, we revealed that DDX21, which exhibited high expression in tumors, showed predominantly high expression in CI compared to CII across 4 CRC databases (Figure 6d,e, *p* < 10^−7^). In addition, EGFR pathway inhibitors showed resistance in the median low DDX21 expression group and enhanced resistance in the high-low DDX21 expression group (Figure 6f). This was further validated in cells with high expression of DDX21 (Figure 6g,h and Appendix A). Consistently, cells with DDX21 were sensitive to the inhibition of CUDU-101, which is a potent multi-targeted inhibitor against HDAC, EGFR, and HER2 pathway (Figure 6i and Appendix A). Moreover, cells with DDX21 high expression exhibited higher sensitivity to DNA replication and genome integrity drugs (Appendix A). The pathway enrichment analysis showed MAPK signaling pathway, DNA replication, and DNA damage repair pathway were enriched in DDX21 high expression group (Figure 6j and Appendix A).

Extensive evidences have demonstrated that CR organoids retain molecular and histological features similar to those of the tumor specimens and faithfully predicts the clinical response of patients [49,50,51,52]. To further reproduce the responses in clinical patients, we established CR (colorectal) tumor-derived organoids. After histological and pathological validation of the organoids (Appendix A), we compared the gefitinib response in organoids with high and low DDX21 expression. As expected, we observed that organoids with higher DDX21 expression were more sensitive to gefitinib (Figure 6k,l). Based on all the evidences above, we concluded that DDX21 could be used as an effective gene to differentiate drug sensitivity in two clusters.

## 3. Discussion

R-loop binding proteins (RLBPs) act extensive essential roles in genome stability. In our study, we profile the RLBP proteomics landscape in colorectal cancer and uncover the associations of the RLBPs with CIN phenotype. We reveal two RLBP clusters with distinct prognoses, molecular features, and TME. In addition, we identify 42 tumor-associated RLBPs and elucidate the functions of those RLBPs on leading differential drug sensitivities.

CIN refers to an abnormal accumulation of gain or loss of entire regions of chromosomes, including intra- and inter-chromosomal aberrations [53]. CIN is considered to promote tumor development by disrupting tumor suppressor genes, increasing the copy number of oncogenes, or altering gene expression [54]. CIN, especially intra-chromosomal aberrations, is the main phenotype of genomic instability in CRC [53]. The diverse dysregulations of genome maintenance, including defects in chromosome segregation, abnormal DNA damage repair, and telomere crisis contribute to CIN. Incorrectly repaired DNA DSBs, the major driver of chromosomal rearrangements, are usually caused by aberrant DNA damage repair systems, such as homologous recombination (HR) and non-homologous end joining (NHEJ) [55,56].

We revealed that CI-pRLBPs were more enriched in the CIN subtype which is characterized by high CNVs. More importantly, we observed a high expression of DNA repair proteins in CI. The homeostatic regulations of DNA damage repair pathways play a fundamental role in maintaining genome integrity [57,58]. The high expression of DNA repair proteins in CI ruins the balance of DNA damage repair. The hyperactivated double-strand repair pathways in CI, including non-homologous end-joining (NHEJ) and homologous recombination (HR) pathways, implicate excessive exchanges of chromosomes, which further account for high CNVs. Moreover, DDX21, which is a significant RLBPs member, was revealed to engage in the HR pathway directly. The high expression of DDX21 leads to excessive sister chromatid exchanges [59]. PARP1 is required for chromosomal translocations; it displaces the Ku complex and initiates an alternative NHEJ pathway [60]. In addition, the pathway enrichment analysis indicated CI with proficient expression of RLBPs was accumulated in transcription and DNA replication, cellular processes which are essential for genome stability.

Recent studies showed that CIN is associated with both poor prognosis and drug resistance [61,62]. Inappropriately, DNA damage repair pathways are expected to be synergically therapeutic strategies for drugs targeting genome stability. This was further validated in the drug sensitivity analysis, as demonstrated by the high percentage of negative correlation of RLBP expression with IC50 values of drugs involved in genome stability (Figure 6). The concept is consistent with our previous findings showing that high expression of RNA helicases, DDX21, renders sensitivity for irinotecan, oxaliplatin, and etoposide [59]. Thus, overall high therapeutic sensitivity in patients would be expected along with better outcomes in CI. Interestingly, we also found CI was more sensitive to genome EGFR inhibitors. EGFR activation is driven by two main pathways: PI3K/AKT/mTOR signal pathway and RAS/RAF/MEK/MAPK signal pathway, which are involved in regulating cell proliferation, survival, cell migration, and angiogenesis [63]. We noticed MAPK and PI3K signaling pathways enriched in CI. The condensed accumulation of MAPK and PI3K genes may attribute to the high growing requirement for CI. Therefore, RLBPs cluster might be regarded as an adequate predictor to evaluate the chemotherapy and targeted therapy. However, the relationship between DNA damage repair and the EGFR signal pathway required further investigations.

CII-dRLBPs contained higher compositions of lymphocytic fractions but exhibited a worse prognosis. Macrophages and progenitor monocytes substantially accumulated in CII. The extensive immunosurveillance factors were preferentially expressed in CII (Figure 4). The comparatively active immune microenvironment may be implied by the lower contribution of DNA damage repair in CII, which leads to an increase in neoantigen expression in tumors. Considering that R-loops and RLBPs play conventional roles in transcription and immune modulation [64], the higher lymphocytic infiltration but lower antigen processing and presentation activity are insufficient to activate the T-cell response and result in a comparatively inactive immune microenvironment. Furthermore, CII could be observed in the consistently high level of TGFβ1 which account for tumorigenesis and metastasis and is associated with poor prognosis [35,65]. By analyzing the correlation between gene expression and the immune cell composition, we found that DKC1 showed a negative correlation with macrophages and central memory T (Tcm) cells, which exhibit long-term memory after activation of immature T-cells by antigens and have the capacity to return to lymph nodes for antigen restimulation. The increased expression of DKC1 with reduced macrophages and CD8+ T-cells infiltration is consistent with the concept that appropriate immune surveillance is required for tumor control. Previous studies also reported RLBPs are involved in immune microenvironment modulation. The high expression of RNaseH1 leads to R-loop degradation, the process which subsequently reduces cytosolic DNA and inhibits type I IFN-dependent rejection of lymphoma cells [66]. Moreover, METTL3/IGF2BP3 inhibition reduces the stabilization of PD-L1 mRNA, thus enhancing anti-tumor immunity through T-cell activation, exhaustion and infiltration [67]. These results implicate that RLBPs engage in inflammatory signaling pathways and immune cell remodeling.

The locations differences of colorectal tumor are correlated with clinical outcomes. LCRC has a better prognosis than right-sided colon cancer (RCRC) in both chemotherapy and targeted treatments [68]. CI contains higher percentage of left-sided colon (LCRC) adenocarcinoma, while CII contained more patient with right-sided colon cancer or mucinous adenocarcinoma (Figure 2). The location variation on the two clusters supports the survival differences, further suggesting the distinct molecular signature of LCRC and RCRC.

In summary, our study delineates associations of RLBPs with CIN in CRC. We constructed RLPBs-based cluster model to distinguish CRC patients and identify their therapeutic utility in chemotherapy and targeted therapy. With the increasing understanding that RLBPs in CIN phenotypes, drug responses and comprehensive roles in shaping the TME, our results provide some new perspectives on using these proteins as therapeutic strategies and preventing therapeutic failure in future clinical applications.

## 4. Materials and Methods

### 4.1. Differential Protein Analysis

A paired Wilcoxon sighed-rank test was employed to detect the difference of protein expression between tumor and paired Para cancerous tissues (Tandem mass tag (TMT) global proteomic analysis of the 100 tumors and 97 paired Para cancerous tissues in 2019 CPTAC CRC, 90 tumors and 30 paired Para cancerous tissues in 2014 CPTAC CRC). Up-regulated or down-regulated proteins in tumors are defined as proteins that are differentially expressed in tumors compared with their matched Para cancerous tissues (log_2_ (fold change (FC)) T/N > 0.5 or < −0.5, *p* < 0.05).

### 4.2. Correlations of Protein to mRNA or Copy Number Alteration (CNA) or DNA Methylation in RLBPs

A total of 204 RLBP genes in the CRC cell lines or tumor were calculated with pairwise Pearson’s correlation coefficients between protein abundance to mRNA or CNA or CpG promotor DNA methylation.

### 4.3. Single-Cell Analyses

The quality control and cell types clustering of CRC Single-Cell Analyses were reproduced from the Smart-seq2 data of CRC single cell databases (GSE146771). The average expression of 42 tumor-associated RLBPs calculated by the Seurat’s AddModuleScore function was defined as the 42 RLBPs score.

### 4.4. Survival Analysis

Kaplan-Meier survival curve (log-rank test) was used to compare the overall survival time (OS) of CRC cancer patients with proteomic subtypes, or high and low expression of RLBPs. OS curve was calculated according to the median cutoff.

### 4.5. NMF Clustering for Proteomic of CRC

Non-negative matrix factorization (NMF) and the R-package (CancerSubtypes, Version: 1.14.0, Taosheng Xu, Shenzhen, China) were employed to identify CRC sample clusters. The 204 RLBPs genes were selected for clustering analysis. The preferred clustering result is determined by using the observed cophenetic correlation between the clusters and the silhouette width of the uniform membership matrix. We also employed the random forest predictor implemented in the R package randomForest to assign the molecular subtypes to each sample based on the protein expression matrix.

### 4.6. Random Forest Model Clustering for Proteomic of CRC

We trained the random forest model with the ‘train’ function in the R caret package. The model parameter was set as: method = ‘rf’, namely using the ‘random forest’ algorithm. Performance estimate parameter was set as: trControl = trainControl (method = ‘cv’, number = 10), namely 10-fold cross-validation (CV), which was implemented as follows: Divide the observations into ten equally sized independent ‘folds’ (each observation appears in only one fold); Hold out one of these folds (1/Kth of the dataset) to use as a test set; Train a model in the remaining K-1 folds; repeat until each of the folds has been held out once; Performance estimate is the average performance across the K held out folds, namely accuracy of the random forest model. In this study, the final value used for the model was mtry = 2 with the largest value of accuracy (0.9105556).

### 4.7. Pathway Enrichment Analysis

Gene Set Enrichment Analysis was performed using the GSEA software (https://www.broadinstitute.org/gsea/; accessed on 15 April 2021). Samples grouped according to the identified subtypes were subjected to GSEA. Molecular Signatures Database (MSigDB) of hallmark gene sets (H), KEGG, Reactome gene sets (C2) and GO gene sets (C5) were used for enrichment analysis, *p*-value of 0.05 was used as a cutoff. The enrichment score (ES) in GSEA was calculated by first ranking the proteins from the most to least significant with respect to the two phenotypes, the entire ranked list was then used to assess how the proteins of each gene set were distributed across the ranked list.

Gene set variation analysis implemented in the GSVA package (version 1.42.0, Robert Castelo, Barcelona, Spain) was used for gene set enrichment analysis. The differences in pathway activities scored per cell line between CI and CII clusters (Appendix A) were calculated with the LIMMA package (version 3.50.3, Gordon Smyth, Canberra, Australian). The reactome pathway analysis and KEGG analysis between two clusters in CRC cells were performed by the clusterProfiler package (version 4.2.2, Guangchuang Yu, Guangzhou, China).

### 4.8. Phosphorylated Proteomic Analysis

POLR2A tandem heptapeptide repeats in the C-terminal domain (CTD) can be highly phosphorylated and the phosphorylation occurs mainly at residues ‘Ser-2′ and ‘Ser-5′ of the heptapeptide repeat. We calculated the total phosphorylation level of ‘Ser-2′ and ‘Ser-5′ in the POLR2A tandem heptapeptide repeats domain of RNA Pol II in each sample, according to the classification of patients. Statistical differences between the two clusters were utilized with Wilcoxon rank-sum test.

### 4.9. Mutation Burden Evaluation

In order to assess the different mutation burden between the two clusters, we calculated the total number of mutations per trillion bases in tumor tissue, including somatic gene coding errors, base substitutions, gene insertions, and the total number of actual errors. We used the Bioconductor R package Maftools to efficiently and comprehensively analyze somatic variants of CRC patients. Then, the Wilcoxon rank-sum test was used to compare the statistic differences.

### 4.10. Comparison of DDR Gene Expression between Two Clusters

The expression of 276 DDR genes in Cluster II relative to Cluster I were calculated and log_2_(Fold change (C II/C I)) > 0.5 or < −0.5, *p* < 0.05 were considered statistically significant.

### 4.11. Immune Score and Stromal Score

Estimation of stromal and immune cells in malignant tumor tissues was based on the Expression Data (ESTIMATE) algorithm (https://bioinformatics.mdanderson.org/estimate/; accessed on 15 April 2021) and Xcell (https://xcell.ucsf.edu/; accessed on 15 April 2021). The degree of infiltration of immune and stromal cells in CRC tissues was calculated by ESTIMATE and Xcell algorithms. Statistical differences between the two clusters were utilized with Wilcoxon rank-sum test.

### 4.12. Comparison of Immune Cell Abundance between Two Clusters

The fraction of immune cell types in a mixed cell population within the leukocyte compartment was estimated using Xcell and CIBERSORT. Indeed, we applied Xcell and CIBERSORT to the 2012 The Cancer Genome Atlas (TCGA) or the 2019 CPTAC CRC RNAseq data, using a set of 64 immune cells (Xcell) or a set of 22 immune cells (CIBERSORT) to determine relative proportions of immune cells in two clusters. Statistical differences between the two clusters were utilized with the Wilcoxon rank-sum test.

### 4.13. Comparison of Immunomodulators between Two Clusters

From published studies, a 78 immunomodulators (IMs) genes list was obtained [35]. We applied the 2012 TCGA CRC mRNA data to calculate differential expression analysis for IMs between two clusters and then performed a Wilcoxon test for the mean expression level of each gene in the two clusters.

### 4.14. Dependency Data

Dependency data were taken from three datasets: Achilles CRISPR genome-wide CRISPR-Cas9 KO data [27], and two other published articles [43,45]. Specifically, the CERES score was used to calculate the dependency score for a given 42 tumor-associated genes.

### 4.15. Association Analysis between RLBPs Protein Expression and Cell Line Drug Response

Cell line drug sensitivity data were downloaded from the Genomics of Drug Sensitivity on the Cancer Project website (GDSC1 and GDSC2). Annotations of compound pathways and targets were also taken from Cancer Cell Line Encyclopedia (CCLE). The correlation between the protein expression of RLBPs individual genes and treatment responses for each drug was determined by Spearman’s rank correlation coefficient.

### 4.16. Cell Viability Measurement

In a 96-well plate, 5000 cells per well were inoculated. The concentration gradient drug treatment with a culture medium was applied on the second day. After being treated with indicated time, cell counting kit-8 reagent (Bimake, Cat#B34304, Houston, TX, USA) and cultural medium were added 1:10 to each well, and plates were incubated at 37 °C for 2 h. Absorbance was measured at 450 nm and normalized against control cells treated with vehicle solution. The experiments were repeated at least three times.

### 4.17. Immunohistochemistry

IHC staining using previously reported methods. Antibodies were optimized with a predetermined staining protocol: MCM3 (Abcam, Cat#ab4460, 1:500, Cambridge, UK), MCM5 (Proteintech, Cat#11703-1-AP, 1:100, Rosemont, IL, USA), CD68 (Santa Cruz, Cat#20060, 1:400, Dallas, TX, USA), DKC1 (Novus, Cat#NBP1-85156, 1:100, Littleton, CO, USA), CD8 (Abcam, Cat#ab199016, 1:500) and observed in Leica Microscope.

### 4.18. Establishment and Culture of PDOs from Colon Cancer

After being washed with iodine tincture and PBS, the biopsies were thoroughly minced into the smallest pieces and digested with 10 mL Gentle Cell Dissociation Reagent (GCDR) (STEMCELL, Cat#07174, Vancouver, BC, Canada) with rotation at 40 rpm for 30 min at 4 °C. Cell pellet was filtered through 70 μm strainer (STEMCELL, Cat#27216) and counted with crypts. Next, 1000 crypts resuspending per dome were plated with Matrigel (Corning, Cat#356231, Corning, NY, USA) at 1:4 into 24 well-pre-warmed cell culture plates. The Matrigel dome was then solidified in a 37 °C incubator for 10 min and cultured with 500 uL IntestiCultTMOGM Human Basal Medium (STEMCELL, Cat#100-0190) mixed with Organoid Supplement (1:1 volume) (STEMCELL, Cat#100-0191), add desired antibiotics 50 ug/mL gentamicin (Thermo Fisher Scientific, Cat#15710064, Waltham, MA, USA) and 100 units penicillin/Streptomycin (Thermo Fisher Scientific, Cat#15140122).

Patient derived organoids (PDOs) were passaged every 5–10 days using TrypLe (Thermo Fisher Scientific, 12604013). In short, 300–600 uL TrypLe were added into the well and thoroughly scrape the Matrigel dome, transferring the organoid mixture to a 15 mL conical tube. Centrifuge the organoid at 290× *g* for 5 min and resuspended in Matrigel and re-seeded at an appropriate ratio. PDOs were cryopreserved with 90% serum + 10% DMSO.

### 4.19. 3D PDOs Drug Screens

Organoids digestion: first, the organs in the petri dish were digested and incubated at 37 °C for 1–5 min. Observed under the microscope every minute, when the organoids were digested into a single cell mass containing 2–3 cells, the medium was added to terminate the digestion, and a 1200 rpm centrifugation for 5 min, the supernatant was discarded, added to 2 mL 2% Matrigel culture medium, and the cell suspension was prepared. Grafted organoids: allowed the 384-well microplates to polymerize. After being cultured in the incubator for two days, the drug was added according to the drug concentration gradient, and then cultured in the incubator for four days. ATP fluorescence detection: after four days of drug culture, the drug sensitivity test was carried out. Chemiluminescence detection was carried out by using a chemiluminescence detector. The tumor inhibition rate and IC50 value of the drug were calculated according to the chemiluminescence values of different concentrations of the drug.

### 4.20. Statistics

The statistical details of all experiments are reported in the form of words, figure legends, and figure diagrams, including statistical analysis, statistical significance, and counting. Data were analyzed with a two-tailed *t*-test and Rstudio software. The correlation between groups was analyzed by Spearman’s or Pearson’s test, and *p* < 0.05 was considered to be statistically significant (* *p* < 0.05; ** *p* < 0.01; *** *p* < 0.001). Statistical analyses were conducted using GraphPad Prism 7 (Harvey Motulsky, California, USA). For all experiments with error bars, the standard error (SEM) of the average was calculated to represent the change in each experiment. The sample size of each experiment and the number of repetitions of the experiment are included in the figure legends.

## 5. Conclusions

We identified two RLBPs subtypes with different therapeutic and prognostic significance in CRC, based on proteogenomic analysis. CI-pRLBPs were associated with CIN, better prognosis, and sensitivity to drugs targeting genome integrity and EGFR.

## Figures and Tables

**Figure 1 cancers-14-05607-f001:**
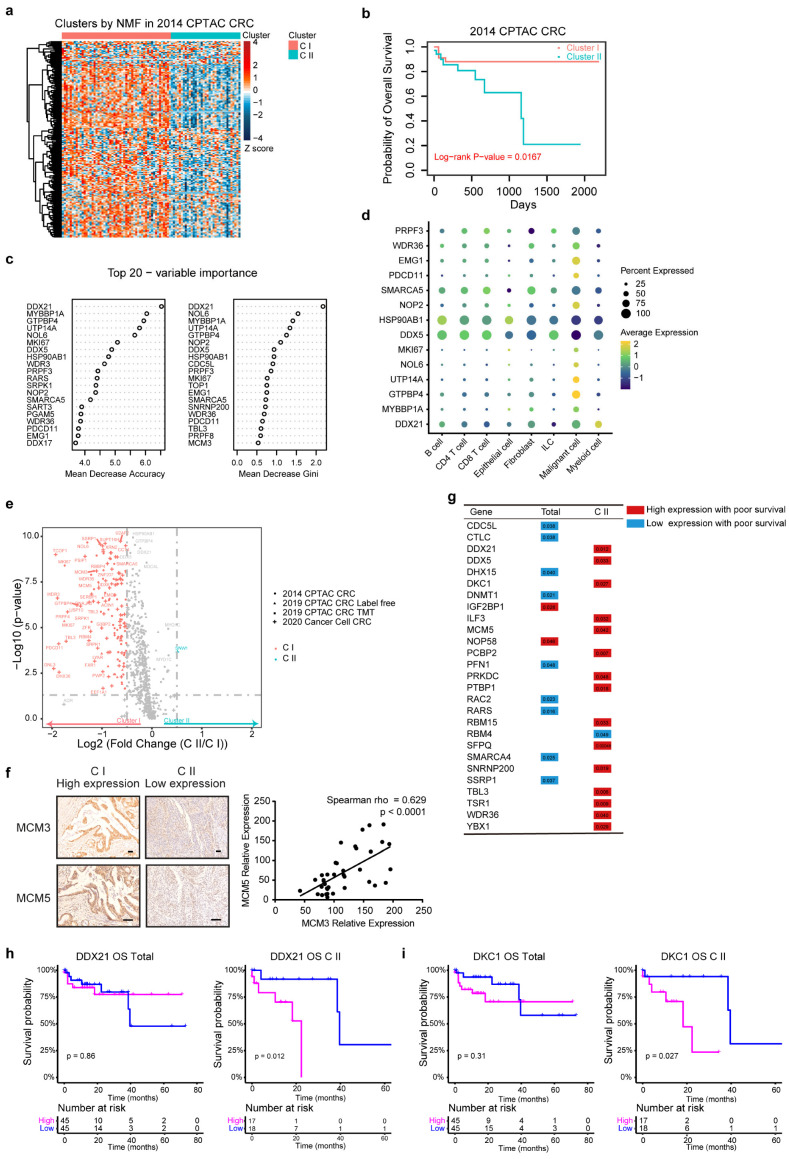
The consistent expressions of RLBPs in two clusters and predictions for prognosis. (**a**) Heatmap illustrating 204 RLBPs expressions in 2014 CPTAC CRC proteomic databases in 90 patients. Annotations indicate tumor clusters CI and CII. (**b**) Kaplan-Meier curves for overall survival (OS) based on two clusters in 2014 CPTAC CRC proteomic databases. (**c**) The top 20 important characteristics indicated by the Mean Decrease Accuracy value and the Mean Decrease Gini value from a random forest model for 2014 CRC databases clustering. (**d**) Bubble heatmap showing the expression of 14 RLBPs in each scRNA−seq cell type. Dot size indicates the percentage of expressed cells, colored by average expression levels. (**e**) Volcano plot showing 204 RLBPs proteomics in four CRC proteomic clusters from three colorectal databases. (Blue and red indicate Log_2_ (fold change (CII/CI) > 0.5 and Log_2_ (fold change (CII/CI) < −0.5 respectively, *p* < 0.05). (**f**) Representative images of MCM3 and MCM5 immunochemistry staining in two clusters of colon cancer tissues (left). Spearman correlation of MCM3 expression with MCM5 expression in colorectal cancer (right). (**g**) Table showing 204 RLBPs that were significantly associated with prognosis in total patients or in CII (Log rank test, *p* < 0.05). (**h**,**i**) Kaplan-Meier curves of overall survival (OS) for total patients or CII with DDX21 (**h**) or DKC1 (**i**) high and low abundance.

**Figure 2 cancers-14-05607-f002:**
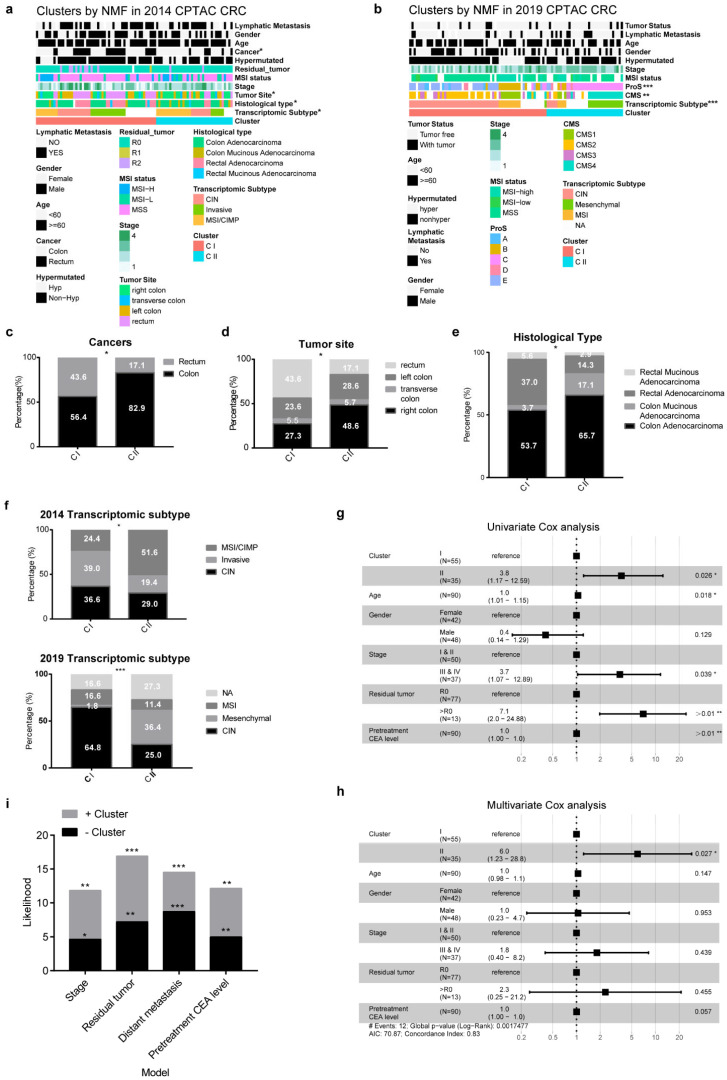
Clinical features in two clusters and Cox regression analysis of colorectal cancer overall survival. (**a**,**b**) Clinical relevance in two clusters in 2014 CPTAC CRC (**a**) and 2019 CPTAC CRC (**b**) proteomic databases. Fisher’s exact test was used for those variables (* *p* < 0.05, ** *p* < 0.01, *** *p* < 0.001). (**c**–**e**) Distributions of cancers (**c**), tumor site (**d**), histological type (**e**) in two clusters in 2014 CPTAC CRC databases. (**f**) Distributions of transcriptomic subtype in two clusters in 2014 or 2019 CPTAC CRC databases. (**g**,**h**) Univariate (**g**) or Multivariate (**h**) Cox regression model analysis, which included the factors of cluster, age, gender, stage, residual tumor, and pretreatment CEA level. (**i**) CoxPH models of stage, residual tumor, distant metastasis, pretreatment CEA level with (Multivariate Cox) or without (Univariate Cox) proteomic Cluster were compared.

**Figure 3 cancers-14-05607-f003:**
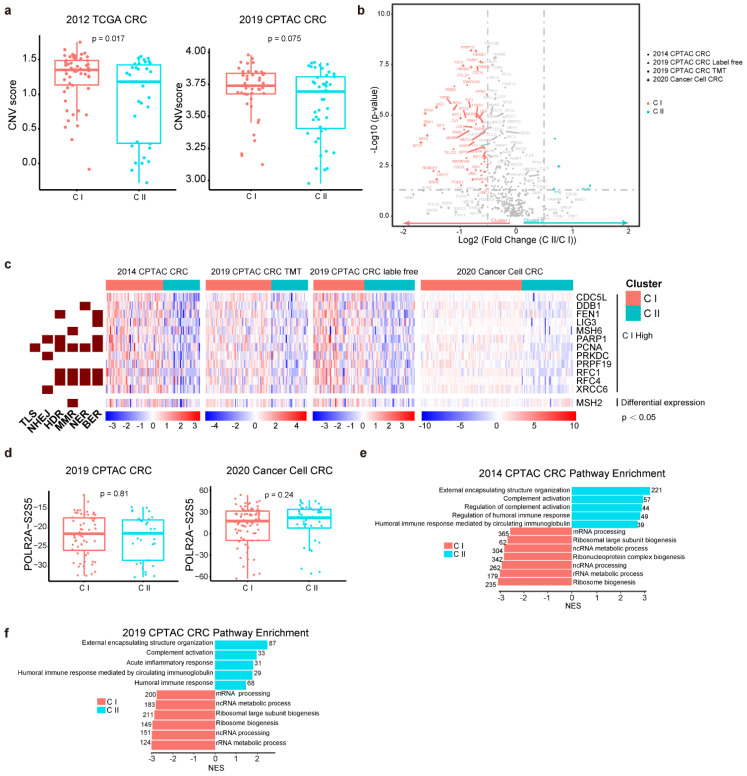
CI and CII possess different molecular characteristics. (**a**) Box plot showing CNV score in two clusters in 2012 TCGA CRC and 2019 CPTAC CRC databases. (**b**) Volcano plot showing expressions of DNA damage proteins in two clusters in three CRC proteomics databases (Blue and red indicate Log_2_ (fold change (CII/CI) > 0.5 and Log_2_ (fold change (CII/CI) < −0.5 respectively, *p* < 0.05). (**c**) Heatmap showing selected DNA damage protein engaged DNA repair pathways (*p* < 0.05) through four proteomic cohorts. (**d**) Box plot showing phosphorylation of Polymerase 2 (PORL2A) at S2/5 sites in two clusters in 2019 CPTAC CRC databases (left) and 2020 Cancer cell CRC databases (right) [23,24]. (**e**,**f**) Pathway enriched in two clusters in two CRC proteomics databases analyzed by GESA (Two−sided unpaired *t* test).

**Figure 4 cancers-14-05607-f004:**
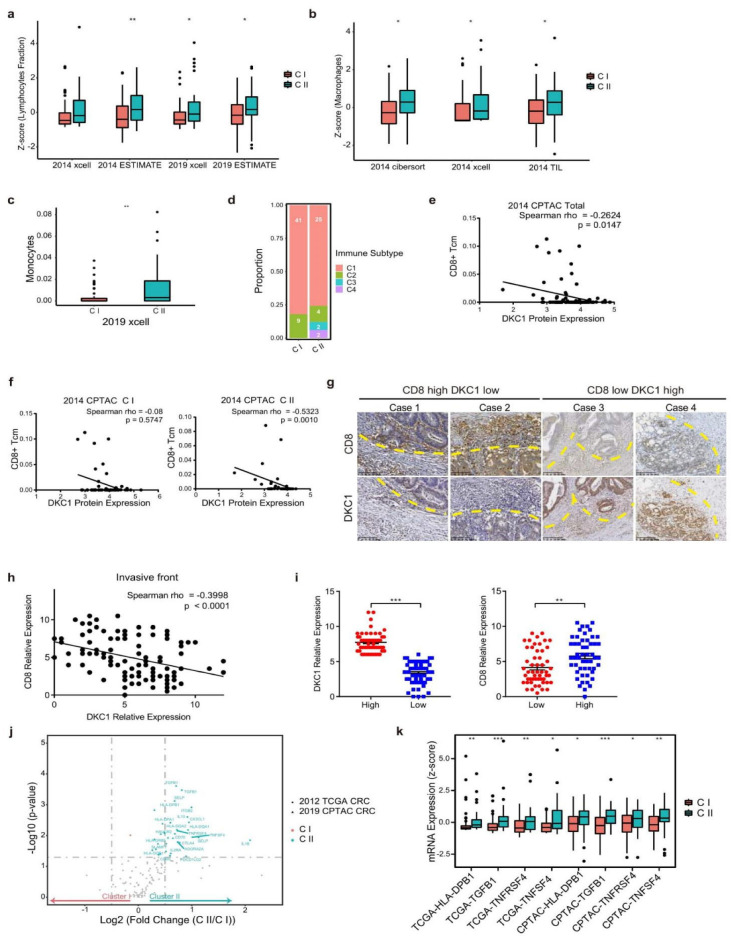
Distinct immune microenvironments exist in CI and CII. (**a**) Comparisons of Lymphocytes Fraction (see Section 4) between two clusters in two CRC databases by different immune faction analysis, xcell, and ESTIMATE [31,32]. (**b**) Comparisons of Macrophages between two clusters in 2014 CPTAC CRC databases analyzed by three methods, xcell, CIBERSORT, and immune staining of CRC databases [31,33,34]. (**c**) Comparisons of Monocytes between two clusters in 2019 CPTAC CRC Label free databases analyzed by xcell. (**d**) The distribution of patients belonging to two clusters in four different immune subtypes from Thorsson et al., 2018, Immunity [35]. (**e**) Scatter diagram showing correlations of DKC1 protein expression with CD8+ Tcm cells in total patients in 2014 CPTAC CRC databases. (**f**) Scatter diagram showing correlations of DKC1 protein expression with CD8+ Tcm cells in CI or CII cluster in 2014 CPTAC CRC databases. (**g**) Representative images of DKC1 immunochemistry staining and CD8+ T cells in colorectal cancer tissues. (**h**) Spearman correlation of DKC1 expression with CD8+ T cells in invasive front (where tumor invaded normal lamina propria) in colorectal cancer. (**i**) The relative DKC1 and CD8 expression of patients in Immunohistochemical high and low score groups (divided by median expression) (* *p* < 0.05, ** *p* < 0.01, *** *p* < 0.001, error bar indicates ±SEM). (**j**) Volcano plot showing immunomodulators’ proteomics in CII versus CI in two CRC proteomics databases. Blue indicates Log_2_ (fold change (CII/CI) > 0.5, *p* < 0.05) [35]. (**k**) The boxplot showing the mRNA expressions of immunomodulators in 2014 and 2019 CRC databases.

**Figure 5 cancers-14-05607-f005:**
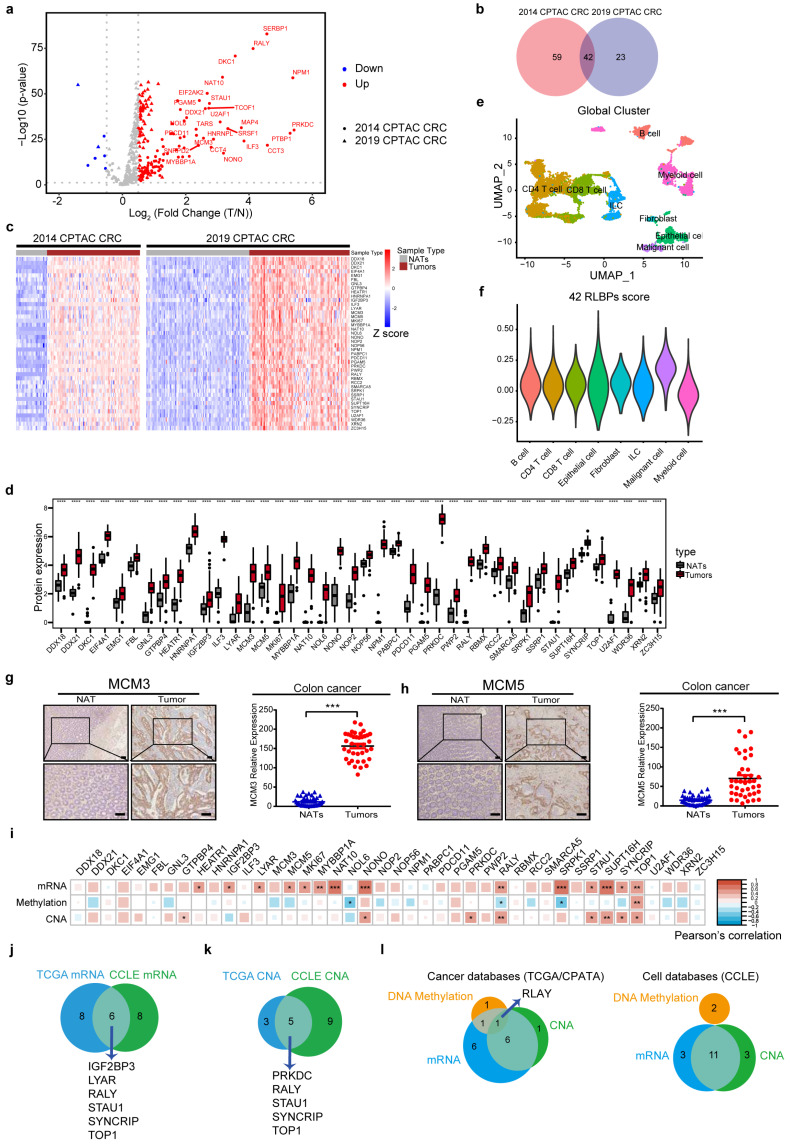
The expression and correlation of 42 RLBPs in colorectal tumors. (**a**) Volcano plot showing 204 RLBPs proteomics in tumors versus normal adjacent tissues (NATs) in two colorectal databases. 2014 CPTAC colorectal cancer (CRC) databases from Bing Zhang et.al., 2014., Nature and 2019 CPTAC colorectal databases from Suhas Vasaikar et.al., 2019, Cell [22,23]. (**b**) Venn diagram depicting tumor high expressing RLBPs in two cancer databases from (**a**) (log_2_ (fold change) > 0.5, *p* < 0.05). (**c**) The heatmap showing RLBPs expression in tumors and NATs in patients from two CRC proteomics databases. (**d**) The boxplot showing RLBPs expression in tumors and NATs in 2014 CPTAC CRC proteomics databases, **** *p* < 0.0001. (**e**) UMAP plots showing major immune and non−immune subsets reproduced by Smart−seq2 scRNA−seq. (**f**) Violin plot showing the 42 RLBPs’ score in all immune and non−immune cell types. (**g**,**h**) Representative images of immunochemistry staining of MCM3 (**e**) or MCM5 (**f**) expressions in colorectal cancer between tumor and NATs. Quantifications of MCM3 and MCM5 expressions in tumors and NATs were shown on right. (**i**) Pearson correlation of RLBPs proteomic expression with CNA, DNA methylation (promoter CpG clusters) or mRNA from TCGA CRC databases. (**j**,**k**) Venn diagram showing genes whose expression was affected by mRNA (**j**) or CNA (**k**) in both CCLE databases and TCGA CRC databases. (* *p* < 0.05, ** *p* < 0.01, *** *p* < 0.001). (**l**) Venn diagrams showing gene expressions affected by CNA, CpG promotor DNA methylations and mRNA in Cancer Cell Line Encyclopedia (CCLE) databases (DepMap Public) [27] or TCGA CRC databases [42] (* *p* < 0.05, ** *p* < 0.01, *** *p* < 0.001, error bar indicates ±SEM).

**Figure 6 cancers-14-05607-f006:**
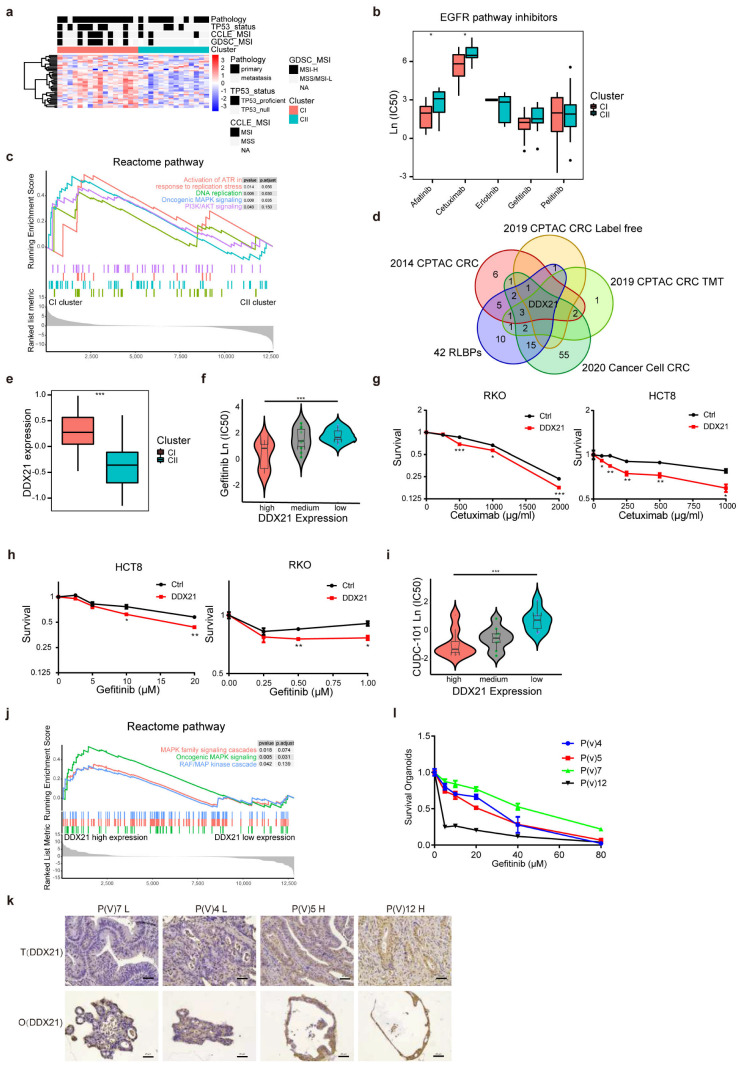
The cell and patient model to verify drug sensitivity. (**a**) Heatmap showing 42 RLBPs expression and clinical features in CRC cells. (**b**) Ln (IC50) of EGFR inhibitors in CRC cells’ sensitivity between two clusters. (**c**) GSEA Reactome enrichment plot showing enrichment pathways in two clusters in CRC cells. (**d**) Venn diagram depicting the overlap genes between tumor high expressed 42 RLBPs and cluster significantly expressed RLBPs in three CRC proteomic databases (*p* < 10 × 10^-7^, Wilcoxon rank−sum test). (**e**) boxplot showing the DDX21 expression in two cell clusters. (**f**) Violin plot showing distributions of Gefitinib’s Ln (IC50) in DDX21 protein high, medium, and low groups (high groups were divided by DDX21 expression higher than 75% cells, low groups were divided by DDX21 expression lower than 25% cells). (**g**) Viability assays of control or DDX21 overexpressing RKO and HCT8 cells treated with Gefitinib (48 h). (* *p* < 0.05, ** *p* < 0.01, *** *p* < 0.001. Error bar indicates ±SEM by Student’s *t*-test). (**h**) Viability assays of control or DDX21 overexpressing RKO and HCT8 cells treated with Cetuximab (96 h). (* *p* < 0.05, ** *p* < 0.01, *** *p* < 0.001. Error bar indicates ±SEM by Student’s *t*-test). (**i**) Violin plot showing distributions of CUDC−101′s Ln (IC50) in DDX21 protein high, medium, and low groups (high groups were divided by DDX21 expression higher than 75% cells, low groups were divided by DDX21 expression lower than 25% cells). (**j**) GSEA Reactome enrichment plot in DDX21 high and low expression groups in CRC cells. (**k**) Immunohistochemistry staining of DDX21 on PODs and corresponding primary tumors. Scale bar, 20 μm. (**l**) Organoids dose−response to gefitinib, surviving organoids data shown are means ± SE from three independent experiments.

## Data Availability

The datasets used in this article for analysis are all public. A 204 DNA/RNA Hybrids gene list was obtained from published studies [16,17]. Source proteomic data, mRNA, mutation, and CNA raw data of three colon cancer in this study can be accessed through the Clinical Proteomic Tumor Analysis Consortium (CPTAC) data portal and from the Firehose website (https://cptac-data-portal.georgetown.edu/study-summary/S045; accessed on 15 April 2021, http://gdac.broadinstitute.org; accessed on 15 April 2021, version 20130523, http://www.biosino.org/node/project/detail/OEP000729; accessed on 15 April 2021). Source proteomic data of Clear Cell Renal Cell Carcinoma, Head-and-neck squamous cell carcinoma, Lung adenocarcinoma, and Hepatocellular Carcinoma can be obtained from the CPTAC data portal (https://cptac-data-portal.georgetown.edu/study-summary/S050; accessed on 15 April 2021, https://cptac-data-portal.georgetown.edu/study-summary/S054; accessed on 15 April 2021, https://cptac-data-portal.georgetown.edu/study-summary/S056; accessed on 15 April 2021, https://cptac-data-portal.georgetown.edu/study-summary/S049; accessed on 15 April 2021). A 276 DDR gene list was obtained from published studies [30]. The gene fitness scores, proteomic data, mRNA expression, and Copy number alteration of the cell lines are available from the project Score web portal: https://score.depmap.sanger.ac.uk; accessed on 15 April 2021. Cancer cell lines’ drug responses from this study were downloaded from the GDSC database. CRC scRNA-seq datasets from human samples can be downloaded from (https://www.ncbi.nlm.nih.gov/geo/query/acc.cgi?acc=GSE146771; accessed on 15 April 2021, GEO accession number GSE146771). The rest of the data available in the article can be found in the Appendix A or from the authors upon request.

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
