# Peer review of "The Association of R-Loop Binding Proteins Subtypes with CIN Implicates Therapeutic Strategies in Colorectal Cancer"

_cancers, 2022, doi:10.3390/cancers14225607_

Round 1

Reviewer 1 Report

This article initiated a comprehensive proteomics analysis on 204 RLBPs in 2014 CPTAC CRC, 2019 CPTAC CRC, and 2020 cancer cell CRC databases. The authors revealed two RLBP clusters with distinct prognosis, CI and CII respectively. The CI clusters correlated with chromosome instability and microsatellite instability phenotypes, and the CII clusters correlated with worst prognosis and low expression of RLBPs. The two clusters were distinguished by physiological features and tumor microenvironment. In addition, they finally identified 42 tumor-related RLBPs, and found that the clusters with high expression of tumor-related RLBPs displayed different sensitivity of drugs involved EGFR and genome stability pathways. Thus, the associations of RLBPs with cancer provided a theoretical basis for further functional investigation of R-loop binding proteins in cancer. However, several questions should be addressed before its publication.

1.     How to define cluster I and cluster II, and what’s the standard of accuracy by the random forest model?

2.     The whole analysis was based on database. The authors should choose at two genes from each cluster to verify their conclusion in CRC patients.

3.     The gene expression in Figure 1f should include CI.

4.     In Figure 2f, it appeared that the percentages of CIN were similar between the CI and CII clusters but the percentages of the invasive and MSI/CIMP subtypes were quite different. Why did the authors focused on CIN?

5.     The results of CNV score from other two databases should be showed in figure 3a. The proteins specifically expressed in CI and CII should be separately showed in figure 3c.

6.     In Figure 3c, more proteins (at least two) representing DNA damage repair event should be included.

7.     What is the molecular mechanism for the functional difference between the CI and CII should be examined.

Reviewer 2 Report

In this report the authors query several public databases to investigate mutation signatures and CINs in colorectal cancers. They perform a battery of statistical analysis and reveal that an increase in CINs correlates with high expression of R-loop binding proteins. Additionally, they also show that CINs correlate with an increase in DNA damage repair genes expression. The study is well presented and warrants publication in Cancers. The results will be of interest to cancer geneticists.

I have one major comment. The observation that CIN frequency correlates with an increase in RLBPs as well as DNA repair genes is not well discussed. A high level of chromosomal re-organization requires an increase in DNA repair genes. This should be discussed and references should be provided. Additionally, what types of CINs are observed in these cancers (inter or intra-chromosomal)? They are not the same as translocations are more likely to drive cancer evolution. A recent review mapped all the CINs in human cancers (PMID 35091282). The authors should discuss their findings in light of some of these findings.

Round 2

Reviewer 1 Report

The authors improved the manuscript and agree it publication.

Reviewer 2 Report

The authors have made significant changes that improved this paper. This reviewer is satisfied.